# Obesity-Related Complications Including Dysglycemia Based on 1-h Post-Load Plasma Glucose in Children and Adolescents Screened before and after COVID-19 Pandemic

**DOI:** 10.3390/nu16152568

**Published:** 2024-08-05

**Authors:** Joanna Smyczyńska, Aleksandra Olejniczak, Paulina Różycka, Aneta Chylińska-Frątczak, Arkadiusz Michalak, Urszula Smyczyńska, Beata Mianowska, Iwona Pietrzak, Agnieszka Szadkowska

**Affiliations:** 1Department of Pediatrics, Diabetology, Endocrinology and Nephrology, Medical University of Lodz, 90-419 Lodz, Poland; joanna.smyczynska@umed.lodz.pl (J.S.); aleksandra.olejniczak@umed.lodz.pl (A.O.); arkadiusz.michalak@umed.lodz.pl (A.M.); beata.mianowska@umed.lodz.pl (B.M.); iwona.pietrzak@umed.lodz.pl (I.P.); 2Department of Pediatrics, Endocrinology, Diabetology and Nephrology, Central Clinical Hospital of Medical University of Lodz, 90-419 Lodz, Poland; prozycka@csk.umed.pl; 3Outpatient Clinic of Diabetology, Central Clinical Hospital of Medical University of Lodz, 90-419 Lodz, Poland; anetamichal@wp.pl; 4Department of Biostatistics and Translational Medicine, Medical University of Lodz, 90-401 Lodz, Poland; urszula.smyczynska@umed.lodz.pl

**Keywords:** simple obesity, oral glucose tolerance test, insulin resistance, type 2 diabetes, prediabetes, intermediate hyperglycemia, metabolic profile

## Abstract

Childhood obesity, with its metabolic complications, is a problem of public health. The International Diabetes Federation (IDF) has recommended glucose levels 1 h post oral glucose load (1h-PG) > 155–209 mg/dL as diagnostic for intermediate hyperglycemia (IH), while >209 mg/dL for type 2 diabetes (T2D). The aim of the study was to assess the occurrence of prediabetes, IH, and T2D in children and adolescents with simple obesity according to the criteria of American Diabetes Association (ADA) and of IDF, and the effect of COVID-19 pandemic on these disorders. Analysis included 263 children with simple obesity, screened either in prepandemic (PRE—113 cases) or post-pandemic period (POST—150 cases). All children underwent 2 h OGTT with measurements of glucose and insulin every 0.5 h, lipid profile, and other tests; indices if insulin resistance (IR): HOMA, QUICKI, Matsuda index, AUC (glu/ins) were calculated. The incidence of T2D, prediabetes, and IH was higher in POST with respect to PRE, with significant differences in the indices of IR, except for HOMA. Significant differences were observed in the assessed parameters of glucose metabolism among the groups with T2D, prediabetes, IH, and normal glucose tolerance (NGT), with some similarities between IH (based on 1h-PG) and prediabetes. Increased frequency of dysglycemia among children and adolescents with simple obesity is observed after COVID-19 pandemic. Metabolic profile of patients with IH at 1h-PG is “intermediate” between NGT and prediabetes.

## 1. Introduction

Childhood obesity is an important problem of public health, especially in economically advanced countries. According to data of the World Health Organization (WHO) [1], the prevalence of overnutrition in children and adolescents aged 5–19 years in a world scale increased from 8% in 1990 to 20% in 2022, including the increase in obesity from 2% to 8%, respectively. Moreover, overnutrition has become a problem of not only high-income countries. As overweight and obesity are related to the imbalance between energy intake (diet) and energy expenditure (mainly physical activity), any situation that favors increase in caloric intake and/or limits daily activity and physical exercise may increase the risk of overnutrition. Undoubtedly, lockdown during COVID-19 pandemic is considered an important factor responsible for creating circumstances conducive to a sedentary lifestyle and unfavorable changes in nutritional habits, leading to the increased incidence of overweight and obesity [2,3,4,5,6]. Also, the impact of chronic stress, related to the pandemic situation, should be taken into account while discussing the causes of increasing incidence of obesity at that time. The latter issue was discussed in a recent paper of Polish authors [7].

Obesity is associated with an increased risk of many complications, including glucose and lipid metabolism disorders. According to the classification of American Diabetes Association (ADA) [8], diagnostic criteria of diabetes include fasting plasma glucose (FPG) ≥ 126 mg/dL or 2 h post oral glucose load (2h-PG) plasma glucose in oral glucose tolerance test (OGTT) ≥ 200 mg/dL, or glycated hemoglobin (HbA1c) ≥ 6.5%, or random plasma glucose ≥ 200 mg/dL in a patient with classic symptoms of hyperglycemia or hyperglycemic crisis. The term “prediabetes” is used for disorders of glucose metabolism that do not meet the criteria of diabetes. It includes two states of intermediate hyperglycemia (IH), i.e., FPG from 100 to 125 mg/dL, that are labeled as impaired fasting glucose (IFG) and 2h-PG plasma glucose from 140 to 199 mg/dL—impaired glucose tolerance (IGT), and/or HbA1c from 5.7% to 6.4% (independently from glucose levels). Prediabetes may be associated with obesity and dyslipidemia, and is considered a risk factor of diabetes and cardiovascular diseases [8]. Apart from current criteria of ADA [8], different definitions of prediabetes and of particular hyperglycemic states have been provided by WHO, ADA, and International Expert Committee (IEC) over the last 50 years [9]. The discrepancies between different diagnostic criteria of diabetes and prediabetes have been widely discussed by Makaroff [10], who has also stressed the need for international consensus on the diagnosis of IH. The author has also noted that not every person with IH would develop diabetes, so the term prediabetes may be misleading.

In children, the role of HbA1c alone in the diagnosis of diabetes is unclear [11]. A previous study conducted by our research group showed significant discrepancies in diagnosing hyperglycemic disorders based on OGTT and the results of HbA1c [12].

According to current guidelines of Diabetes Poland [11], screening for diabetes should be performed—among others—in adult persons with overweight and obesity, as well as in children over 10 years of age or after puberty with BMI ≥ 85th centile for age and sex, and risk factors of T2D. This statement is in line with the recommendations of ADA [8].

In 2024, the International Diabetes Federation (IDF) has recommended that glucose levels 1 h post oral glucose load (1h-PG) should be included in OGTT interpretation, with values > 155–209 mg/dL considered diagnostic for IH, while >209 mg/dL for T2D [13]. Thus, in the patients with normal glucose tolerance (NGT), according to the criteria of ADA [8] (i.e., FPG < 100 mg/dL, 2h-PG glucose < 140 mg/dL, and HbA1c < 5.7%), IH should be diagnosed in the case of 1h-PG plasma glucose concentration > 155–209 mg/dL. In all the states of IH, the possibility of progression to T2D should be taken into account. Accordingly, the diagnosis of T2D in the modification proposed by IDF [13] should include the standard criteria of ADA [8] and/or 1h-PG plasma glucose over 209 mg/dL.

There have been only few studies implementing the IDF diagnostic criteria in a pediatric population. The significance of measuring 1h-PG glucose levels in children and youths with cystic fibrosis has been documented in the context of developing diabetes in future [14,15]. In children with obesity, elevated 1h-PG glucose have been associated with obstructive sleep apnea [16]. To our best knowledge, no results of studies comparing carbohydrate metabolism disorders in children before and after the COVID-19 pandemic, taking into account the IDF classification, have been published so far.

Apart from direct interpretation of glucose and insulin concentrations at particular time points of OGTT, some indices of insulin sensitivity (IS)/insulin resistance (IR) were introduced. It seems worth mentioning that different indices are considered to have different meanings, e.g., homeostasis model assessment (HOMA) is an index of hepatic IR, quantitative insulin sensitivity check index (QUICKI), and Matsuda index—of IS, while AUC (ins/glu)—of beta-cell function [17,18,19].

The aim of the study was to assess the occurrence of carbohydrate metabolism disorders and to compare selected indices of IS/IR and of other metabolic complications in children and adolescents with simple obesity, taking into account the diagnostic criteria of prediabetes, IH, and T2D proposed by ADA and IDF [8,13], as well as the possible effect of COVID-19 pandemic on these disorders.

## 2. Materials and Methods

The retrospective, observational study included two cohorts of children and adolescents, age 6–18 years (hereinafter referred to as “children”), with simple obesity, screened either in prepandemic (Group PRE) or post-pandemic (Group POST) period. Group PRE consisted of 113 children (59 boys, 54 girls), diagnosed in one center in Poland from 2013–2016, while group POST included 150 children (75 boys, 75 girls), diagnosed in the same center from the beginning of 2022 to the end of June 2024. In all patients, height was measured with the accuracy of 0.5 cm and body weight was measured on an electronic scale with the accuracy of 0.1 kg. Next, body mass index (BMI) was calculated and expressed as BMI z-score for age and sex, according to reference data for Polish children [20,21]. Diagnostic criteria for simple obesity were BMI z-score > 2.0, confirmed hyperalimentation, excluded causes of secondary obesity (genetic syndromes, endocrinopathies, glucocorticoid administration). The patients with simple obesity as the only identified health problem were included, while exclusion criteria were secondary obesity, confirmed or suspected genetic syndromes, hypothyroidism, Cushing syndrome and other disorders of endocrine glands or hormonal treatments, previously diagnosed type 1 diabetes, as well as chronic diseases and acute inflammatory disorders. All children had performed OGTT with glucose dose 1.75 g/kg, not exceeding 75.0 g. During the test, glucose and insulin concentrations were measured at 0 (FPG), 0.5 h (0.5h-PG), 1 h (1h-PG), 1.5 h (1.5h-PG), and 2 h (2h-PG). Next, indices of insulin sensitivity (IS): HOMA, QUICKI, Matsuda index, and AUC (ins/glu) were calculated according to the following equations:HOMA = FPG [mg/dL] × fasting insulin [mIU/L]/405;QUICKI = 1/log fasting insulin [mIU/L] + log FPG [mg/dL];Matsuda index = √fasting insulin [mIU/L] × FPG [mg/dL] × mean OGTT glucose [mg/dL] × mean OGTT insulin [mUI/L];AUC (ins/glu) = AUC-ins in OGTT/AUC-glu in OGTT (timepoints 0, 0.5h-PG, 1h-PG, 1.5h-PG, 2.0h-PG were included, calculation according to trapezoidal rule).

Simultaneously, in the first time point of OGTT (0 min), before glucose load, HbA1c percentage, total cholesterol (T-Ch), HDL- and LDL-cholesterol fractions, triglycerides (TG), uric acid, creatinine, and transaminases (ALT, AST) concentrations were measured. In all the patients, diagnostic tests were performed in morning hours, after the night’s rest; children were fasting at least 10 h, they did not receive medications that could affect carbohydrate metabolism nor did they follow any specific diet.

The obtained results were compared in the groups of patients diagnosed before and after the COVID-19 pandemic. Next, analysis was performed after dividing the patients into three groups, according to the diagnostic criteria of ADA for T2D (FPG ≥ 126 mg/dL or 2h-PG glucose ≥ 200 mg/dL, HbA1c ≥ 6.5%), prediabetes (FPG 100–125 mg/dL, 2h-PG glucose 140–199 mg/dL, HbA1c 5.7–6.4%), and NGT (FPG < 100 mg/dL or 2h-PG glucose < 140 mg/dL and HbA1c < 5.7%) [8]. Finally, the patients were reclassified with respect to the glucose concentration at 1h-PG, with respect to the cut-off values for IH and T2D proposed by IDF (>155–209 mg/dL and >209 mg/dL, respectively). Thus, the following groups of patients were created:T2D-IDF—patients fulfilling diagnostic criteria of T2D according to IDF—patients with T2D according to ADA and patients with 1h-PG plasma glucose > 209 mg/dL;Prediabetes-IDF—patients with IFG and/or IGT according to ADA, and/or HbA1c 5.7–6.4% (except for ones, qualified to T2D-IDF group);IH-1h—patients with NGT according to ADA and 1h-PG plasma glucose > 155–209 mg/dL;NGT-IDF—patients with NGT according to ADA and 1h-PG plasma glucose ≤ 155 mg/dL.

Without questioning the comments of Makaroff [10], concerning the differences between IH and prediabetes, in this part of the study, the term “prediabetes” was used for all hyperglycemic states not fulfilling the criteria of T2D, except for IH-1h, according to IDF [13].

Plasma glucose and insulin concentrations were measured with an Allinity Abbott analyzer. HbA1c was assessed with the accuracy of 0.1%, using high-performance liquid chromatography method, certified by National Glycohemoglobin Standardization Program. Biochemical tests (lipid profile, uric acid, creatinine, ALT, AST) were performed with Allinity Abbott analyzer.

Distribution of particular variables within the groups was assessed with Kolmogorov–Smirnov test. As normal distribution was not maintained for the majority of variables, the results were presented as median (interquartile range), and nonparametric tests were used—Mann–Whitney U-test for comparisons between two groups and Kruskal–Wallis ANOVA with appropriate post hoc comparisons with Dunn test with for more than two independent groups. Differences were considered significant at *p* < 0.05. All analyses were performed using T_IBCO_ Statistica 13.1 (Santa Clara, CA, United States).

## 3. Results

Auxological characteristics of the whole study group are presented in Table 1, and indices of IS, HbA1c levels, and results of other biochemical tests are presented in Table 2. For details concerning the results of OGTT, see Appendix A.

### 3.1. Comparisons between the Patients Diagnosed before and after COVID-19 Pandemic

Auxological characteristics of groups PRE and POST are presented in Table 1, results of OGTT are shown in Figure 1 (for details, see also Appendix A), and indices of IS, HbA1c levels, and results of other biochemical tests are shown in Table 2.

There was no difference between the groups PRE and POST in age, height, and weight; nevertheless, BMI was higher in POST than in PRE, with the difference in BMI z-score close to significance (*p* = 0.056). Significant differences were observed in fasting plasma glucose and insulin concentrations between PRE and POST (*p* < 0.001), with the same related to glucose in 1h-PG (*p* = 0.024) and insulin in 0.5h-PG (*p* = 0.031). There was no significant difference in AUC-glu and AUC-ins between the groups; nevertheless, AUC (ins/glu) was significantly lower in POST than in PRE. There was also no significant difference in HOMA between PRE and POST groups; however, both QUICKI and Matsuda index were higher in POST than in PRE (*p*= 0.006 and *p* = 0.044, respectively). The only difference in lipid profile was that in T-Ch, which was significantly higher in Group PRE than in POST (*p* = 0.026), with no difference in the assessed parameters of liver and kidneys function.

According to the criteria of ADA [8], T2D was diagnosed in 7 patients (1 in group PRE and 6 in group POST), prediabetes—in 74 patients (31 in group PRE and 43 in group POST), while NGT—in the remaining 182 patients (81 in group PRE and 101 in group POST) (see Figure 2).

According to the criteria of IDF [13], T2D was diagnosed in 11 patients (2 in group PRE and 9 in group POST), prediabetes—in 71 patients (31 in group PRE and 40 in group POST), IH-1h—in 37 patients (13 in group PRE and 24 in group POST), and NGT (no identified disorders)—in 144 patients (67 in group PRE and 77 in group POST) (see Figure 3).

Incidence of T2D was higher in POST than in PRE, however insignificantly (probably due to small number of patients with T2D). Nevertheless, a difference in distribution of particular diagnoses according to ADA between the groups PRE and POST, assessed in exact Fisher test for multiple groups, was significant (*p* < 0.001).

### 3.2. Comparisons between the Groups of Patients Diagnosed According to the Criteria of ADA [8]

Auxological characteristics of the groups T2D, prediabetes, and NGT, divided according to the criteria of ADA [8], are presented in Table 3, results of OGTT in Figure 4 (for details see also Appendix A), indices of IS, HbA1c levels, and the results of other biochemical tests in Table 4.

There was no significant difference in age, height, BMI, and BMI z-score between the groups with T2D, prediabetes, and NGT. Apart from the differences in glucose concentrations, related mainly to the diagnostic criteria for particular groups, significant differences were observed among these groups in insulin concentrations during OGTT, indices of IR (HOMA, QUICKI, and Matsuda) T-Ch, LDL, and TG, as well as in the concentrations of uric acid. Significant differences were observed in insulin concentrations at timepoint 0, 1h-PG, 1.5h-PG, and 2h-PG between the groups prediabetes and NGT, with higher insulin concentrations in the prediabetes group (see Figure 4). A delayed insulin surge in the T2D group should be noted. There were also significant differences (*p* < 0.001) of AUC-glu between all groups, except for between the groups T2D and prediabetes (*p* = 0.51). AUC-ins was the highest in prediabetes, while the lowest in T2D but with a significant difference only between NGT and prediabetes. The difference in AUC (ins/glu) in the Kruskal–Wallis test was close to significance (*p* = 0.054). It should be noted that the value of lower quartile of AUC (ins/glu) index was relatively low in T2D, which may reflect impaired insulin secretion in some patients from this group. The NGT group differed significantly from both other groups in HOMA, QUICKI, Matsuda index, with the highest HOMA and HbA1c, and the lowest QUICKI and Matsuda index in T2D, and the lowest HOMA and HbA1c, and the highest QUICKI and Matsuda index in NGT. In lipid profile, T-Ch levels were significantly higher in T2D than in NGT; there were also significant differences in LDL and in TG between NGT and both other groups.

### 3.3. Comparisons between the Groups of Patients Diagnosed According to the Criteria of IDF [13]

Auxological characteristics of particular groups of patients, divided according to the criteria of IDF, are presented in Table 5, results of OGTT are shown in Figure 5 (for details see also Appendix A), and indices of IS, HbA1c levels, and results of other biochemical tests are shown in Table 6.

There was no significant difference in age, height, BMI, and BMI z-score between the groups diagnosed according to the criteria of IDF [13]. Significant differences in FPG were observed between T2D-IDF and both IH-1h and NGT-IDF, as well as between prediabetes-IDF and NGT-IDF but not between IH-1h and both prediabetes-IDF and NGT-IDF. At 0.5h-PG, 1h-PG, and 1.5h-PG, glucose concentrations were significantly lower in NGT-IDF than in all the remaining groups, with no difference between T2D-IDF, prediabetes-IDF, and IH-1h (except for a significant difference between T2D-IDF and IH-1h at 1.5h-PG). At 2h-PG, all the differences in glucose levels among the groups were significant, except for that between IH-1h and NGT-IDF (for details, see Figure 5a). A significant difference in fasting insulin concentration was observed between prediabetes-IDF and IH-1h (with the lowest fasting insulin concentrations among all groups in IH-1h). In the remaining time points of OGTT, insulin levels in IH-1h were significantly higher than in both prediabetes-IDF and NGT at 1h-PG, while significantly lower in 2h-PG, while there was no difference between T2D-IDF and prediabetes-IDF at any time point (for details, see Figure 5b). There were also significant differences (*p* < 0.001) of AUC-glu between Norm-IDF and the remaining groups, but not between T2D-IDF, prediabetes-IDF, and IH-1h. Median AUC-ins was higher and very similar in T2D-IDF and prediabetes-IDF groups, while lower and also similar in IH-1h and NGT-IDF groups; however, the difference was significant (*p* < 0.001) only between norm and prediabetes-IDF. There was no significant difference in AUC (ins/glu) index among the groups.

The differences In HOMA and QUICKI between particular groups In post hoc tests were only close to significance. Significant differences between the groups were observed for Matsuda index (*p*-values ranged from <0.001 to 0.020 for particular post hoc comparisons), except for those between T2D-IDF and prediabetes-IDF, and between IH-1h and NGT-IDF (with very similar median value of Matsuda index in the latter groups). The differences in HbA1c were significant between prediabetes-IDF and NGT-IDF (*p* < 0.001), between prediabetes-IDF and IH-1h (*p* = 0.008), and between T2D-IDF and NGT-IDF (*p* = 0.01), but not for T2D-IDF and both prediabetes-IDF and IH-1h, as well as for NGT-IDF and IH-1h.

In post hoc tests, there was no significant difference in T-Ch, LDL, and HDL concentrations among the groups, while significant differences were observed in TG levels, between T2D-IDF and both NGT-IDF (*p* = 0.04) and IH-1h (*p* = 0.03), as well as between prediabetes-IDF and NGT-IDF (*p* = 0.03), with no significant difference between T2D-IDF and prediabetes-IDF, and the same median TG concentrations in NGT-IDF and IH-1h.

### 3.4. Characteristics of the Patients with T2D Diagnosed According to the Criteria of ADA and IDF

The results obtained for the groups T2D according to ADA and T2D-IDF, especially lack of significant differences with other groups, should be interpreted with caution, due to the relatively small number of patients in these groups. For the same reason, statistical analysis of differences between the patients who fulfill the diagnostic criteria of T2D according to ADA or only according to IDF is not possible. Nevertheless, some differences in the results of OGTT seem noteworthy. Glucose and insulin concentrations in OGTT for particular patients are shown in Figure 6. It should be noted that some patients from the T2D-ADA group were diagnosed due to increased HbA1c but not glucose levels. Patients from the T2D-IDF group seem to have more structured and unified shapes of glucose and insulin secretion profiles during OGTT than ones with T2D-ADA; nevertheless, the group size is too small for any far-reaching conclusions.

### 3.5. Characteristics of the Patients with T2D Diagnosed According to the Criteria of ADA and IDF

Significant (*p* < 0.05) but not strong correlations were observed between patients’ age and BMI z-score (r = 0.33), fasting insulin (r = 0.15), HOMA-IR (r = 0.17), QUICKI (r = −0.23), Matsuda index (r = −0.19), HDL (r = −0.16), and uric acid (r = 0.21), speaking for the increase in obesity, IR, and other metabolic disorders with age.

There were also significant (*p* < 0.05) but rather weak correlations between BMI z-score and fasting insulin (r = 0.15), HOMA-IR (r = 0.18), HDL (r = −0.13), HDL (r = −0.18), and uric acid (r = 0.17), to some extent confirming the relationships between the degree of obesity and metabolic disorders.

## 4. Discussion

The study is a continuation of previous research of our study group conducted in the prepandemic period [12], however, with more restricted diagnostic criteria of obesity applied at present. In that former study, discrepancies between hyperglycemic states diagnosed on the basis of OGTT or HbA1c were documented.

In the current study, we did not find clinically important differences in glucose and insulin concentrations during OGTT, lipid profile, and selected parameters of liver and kidneys function between the children with simple obesity, diagnosed before (PRE) or after (POST) the COVID-19 pandemic, even though single differences reached the level of statistical significance. However, significant differences were found in the majority of indices of IR (except for HOMA). As particular indices of IR are considered as markers of different disorders [17], the obtained results may indicate the differences in beta-cell function (AUC (ins/glu)) and peripheral insulin sensitivity (Matsuda index), while not in hepatic insulin resistance (HOMA). Therefore, these data give some evidence for the effect of COVID-19 pandemic on the development of metabolic disorders related to obesity. It should be mentioned that the aim of the present study was to compare only the groups of children with simple obesity, qualified by diagnostics during hospitalization, mainly due to extreme obesity and/or the lack of effects of therapy in outpatient clinic; thus, the presented results do not illustrate the incidence of obesity-related disorders at the population level. In addition, the higher incidence of T2D in the patients diagnosed after the COVID-19 pandemic and differences in distribution of particular diagnoses according to ADA seem worth noting.

Increased rates of T2D in the pediatric population during the COVID-19 pandemic have also been reported by other authors [22,23]. The possible explanations for this finding have included the rise of obesity rates; however, the need for further studies on the relationships between the COVID-19 pandemic and DM2 in youth has been stressed.

Our observations stay in line with previous studies conducted over the world, related to the effect of the COVID-19 pandemic on increased rate of childhood obesity with its metabolic consequences, considered even as “the second pandemic” or “posttraumatic stress disorder”. The authors from different countries have pointed to unfavorable changes in nutritional habits, restricted physical activity, and stress-related psychological problems concerning both children and their caregivers [3,4,5,6,7,24,25,26,27,28,29,30,31,32,33,34,35,36].

On the other hand, it is well known that people with obesity are more susceptible to the complications of viral infections. Recent years have provided evidence that obesity itself, obesity-related complications, and diabetes mellitus worsened the clinical course of COVID-19 infections, both in adults [37,38,39,40,41,42,43,44,45] and in children [46]; however, conflicting results in critically ill adults have also been published [47]. Upregulation of inflammatory genes and related pathways has been documented to be the link between obesity and developing severe COVID-19 [48]. Also, nondiabetic hyperglycemia has been identified as a risk factor of increased COVID-19-related mortality in adults [49].

Despite the fact that, at present, COVID-19 seems to be controlled due to widescale vaccinations, there is a potential risk of further pandemics in the future. In this aspect, implementing the programs of lifestyle and diet modifications for weight control in different age groups seems to be of particular importance. Such interventions are undertaken in many countries from different regions of the world [32,50,51,52,53]; others report the urgent need to reduce the incidence of overweight and obesity [54]. These aims may be achieved by implementing appropriate diet (change in eating habits) and increasing physical activity of affected persons, with important roles of supportive psychological interventions [55,56]. The need for a personalized approach, taking into account sociodemographic, physical, and mental health characteristics of populations covered by the interventions, has also been stressed [57]. It seems that in the current situation of the “obesity pandemic” after the COVID-19 pandemic, they should be treated as a priority in healthcare systems.

In the context of our study, the significance of 1h-PG glucose concentration in interpreting OGTT with respect to distant and long-term complications, including cardiovascular diseases and all-cause mortality, has been well documented in adults (see [13]). Nevertheless, such complications are not observed in childhood. Therefore, it is very difficult or even impossible to perform the studies with similar endpoints on a pediatric population.

In the context of our study, the significance of 1h-PG glucose concentration in interpreting OGTT with respect to distant and long-term complications, including cardiovascular diseases and all-cause mortality, has been well documented in adults (see [13]). Nevertheless, such complications are not observed in childhood. Therefore, it would be very difficult or even impossible to perform the studies with similar endpoints in a pediatric population.

After dividing the patients according to the diagnostic criteria of ADA [8], significant differences in the results of OGTT were observed between the groups with T2D, prediabetes, and NGT, with the highest glucose concentrations and the lowest insulin concentrations in T2D, except for considerable insulin surge at 2h-PG. Higher insulin levels in the prediabetes group than in the T2D group at the remaining time points of OGTT seem to illustrate the pathogenesis of T2D, which involves a transition from hyperinsulinemia caused by IR to relative and then absolute deficit of insulin. However, most pronounced IR was observed in the T2D group.

It is well known that FPG and HbA1c may fail to identify some patients at early stages of glucose metabolism dysregulation. In a quite recent paper of Lechner et al. [58], Matsuda index was described as a good predictor of IR. In our study group, there was a difference in Matsuda index between the groups diagnosed with T2D or prediabetes and the groups diagnosed with IH-1H or NGT according to the criteria of IDF [13]. This observation points to the discrepancies between the results of different tests, without indicating which test has greater diagnostic value. Publications on the significance of glucose concentrations measurement during OGTT at 1h-PG in children are few. Nevertheless, it has been quite recently documented that 1h-PG glucose concentration together with HbA1c is an effective predictor of diabetes in children with obesity or overweight [59]. Similarly, Kostopoulou et al. [60] documented the importance of glucose and insulin concentrations during OGTT, not only at timepoint 0 (fasting) and at 2h-PG.

It seems that dividing the patients with NGT according to the criteria of ADA [8] with respect to glucose concentration at 1H-PG into the subgroups IH-1h and NGT, as proposed by IDF, may be useful for identifying some disorders that could be overlooked during standard 2-time-point OGTT. However, it should be noted that in our study, the group of children fulfilling the criteria of IH-1h is relatively large, so keeping such patients under close surveillance at the population level may pose a challenge to the healthcare system.

Apart from T2D, hyperglycemic states and IR are known risk factors of hepatic steatosis and cardiovascular diseases. This issue was not analyzed in detail in our study; nevertheless, we observed differences in ALT levels among the groups divided according to the IDF criteria [13], as well as some relationships between the severity of disorders of carbohydrate and lipid metabolism. Significant differences in the values of ALT, glucose at 2h-PH, fasting insulin, and indices of IR between obese children with and without nonalcoholic fatty liver disease have been documented by Polish authors [61]. It should be taken into account that metabolic disorders in obese patients progress with age and may be absent or “covert” during childhood. Thus, the results obtained in children may be different from these reported for adults, especially not showing all the complications that would develop with age. Additionally, our study group included only the patients with obesity that might be considered a kind of bias, making impossible assessment of the analyzed parameters in children with different nutritional status. The undoubted limitation of our study is a lack of control group of healthy, nonobese children. Extended diagnostics of such a group are difficult for ethical reasons and even impossible in observational, noninterventional retrospective studies.

The authors are aware that the retrospective design of the study may be a source of potential biases; however, it would be impossible to apply the new diagnostic criteria in a prospective study to a group of patients diagnosed before the COVID-19 pandemic. Moreover, in a retrospective study, we were not able to collect complete and reliable data concerning diet, physical activity, or socioeconomic status of the children; therefore, these otherwise potentially important cofounding factors could not be included in the analysis.

The issue of optimal diagnostic criteria of T2D seems to be very interesting, as some differences are observed in glucose and insulin secretion profiles between the patients diagnosed with respect to different criteria (ADA vs. IDF). Unfortunately, due to the small group size, there was no opportunity to conduct a reliable, more detailed statistical analysis.

Recent research on the issues related to IR and T2D development attempts to identify candidate genes associated with polygenic multifactorial T2D [17] and creating new models of IS and insulin secretion (beta-cell function) [62]. A computational approach may be useful to determine diagnostic accuracy and the optimal cut-offs for different indices of IR, as documented in adult men [63]. These directions should be included in further studies in children.

## 5. Conclusions

An increase in the frequency of hyperglycemic disorders, including T2D, among children and adolescents with simple obesity is observed in the first post-pandemic COVID-19 years with respect to the prepandemic period. 

Among children with simple obesity, the metabolic profile of patients with IH at 1h-PG, according to the criteria of IDF, as the only disorder of glucose metabolism is—in general—“intermediate” between those of ones with NGT and ones with prediabetes, diagnosed with respect to the same criteria [13]. Some similarities of patients with IH at 1h-PG to ones with prediabetes and differences with respect to ones with NGT may reflect the earliest disorders of insulin secretion and/or sensitivity; however, confirming this hypothesis requires continuation of research, including follow-up of the patients with NGT according to the criteria of ADA [8] and IH at 1h-PG according to IDF [13]. The obtained results indicate the usefulness of glucose concentration measurement at 1h-PG in children with simple obesity for detecting disorders that do not meet standard criteria of prediabetes. Assessment of the significance of diagnosing T2D in children and adolescents according to the criteria proposed by IDF [13] requires performing analysis on a definitely larger group of patients.

Further long-term prospective studies including larger groups of obese and nonobese children and adolescents with different diagnoses related to the disorders of glucose metabolism seem necessary to clarify the problems identified and highlighted in the current paper.

## Figures and Tables

**Figure 1 nutrients-16-02568-f001:**
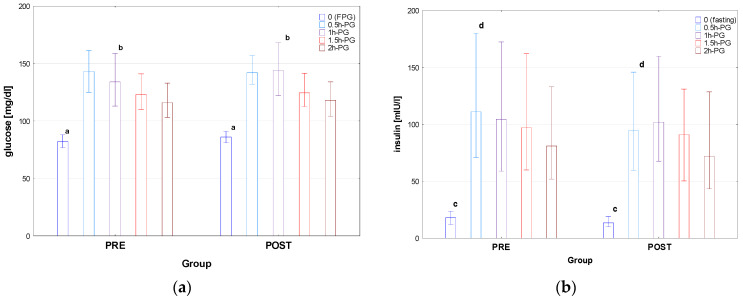
Glucose (**a**) and insulin (**b**) concentrations during OGTT in the patients diagnosed before (PRE) and after (POST) the COVID-19 pandemic. Glucose and insulin concentrations are expressed as median values (column) and interquartile range (whiskers). Significant differences: a, c—*p* < 0.001, b—*p* = 0.024, d—*p* = 0.031.

**Figure 2 nutrients-16-02568-f002:**
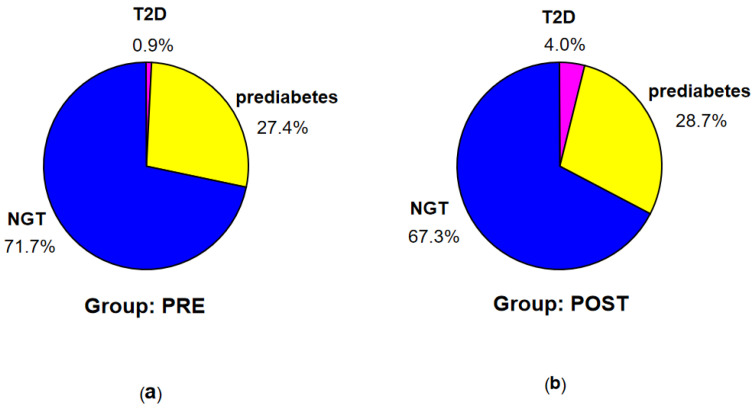
Percentage of the patients with T2D, prediabetes, and NGT according to the criteria of ADA in groups PRE (**a**) and POST (**b**).

**Figure 3 nutrients-16-02568-f003:**
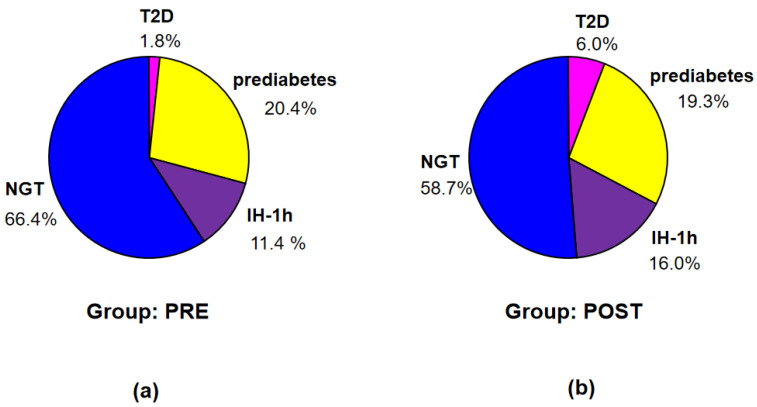
Percentage of the patients with T2D, prediabetes, IH-1h, and NGT according to the criteria of IDF in groups PRE (**a**) and POST (**b**).

**Figure 4 nutrients-16-02568-f004:**
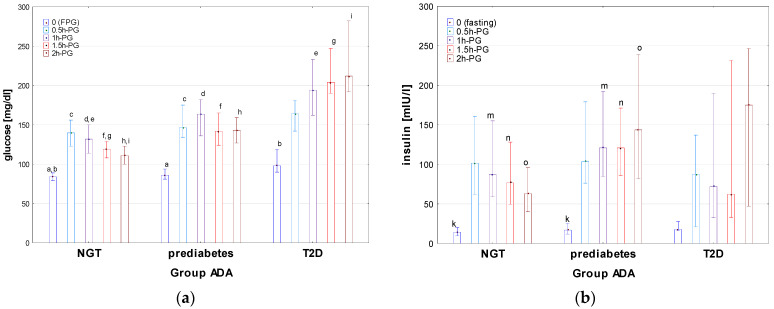
Glucose (**a**) and insulin (**b**) concentrations during OGTT in the patients with T2D, prediabetes, and NGT, diagnosed according to the criteria of ADA. Glucose and insulin concentrations are expressed as median values (column) and interquartile range (whiskers). Significant differences in post hoc tests: a, b—*p* = 0.007, c—*p* = 0.002, d–i—*p* = <0.001, k—*p* = 0.04, m—*p* = 0.004, n—*p* < 0.001.

**Figure 5 nutrients-16-02568-f005:**
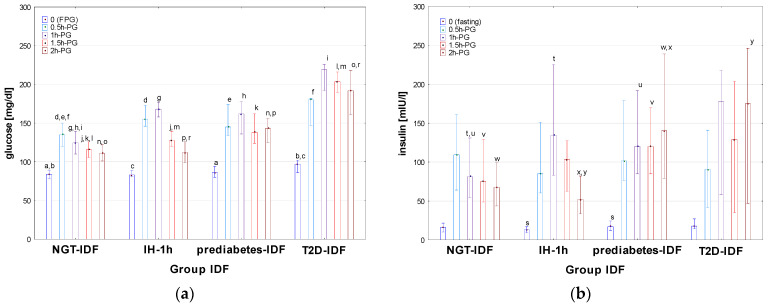
Glucose (**a**) and insulin (**b**) concentrations during OGTT in the patients diagnosed with T2D, prediabetes, IH-1h, and NGT according to the criteria of IDF. Glucose and insulin concentrations are expressed as median values (column) and interquartile range (whiskers). Significant differences in post hoc tests: a—*p* = 0.03, b—*p* = 0.002, c—*p* = 0.02, d–l—*p* < 0.001, m—*p* = 0.005, n–p,r—*p* < 0.001, s—*p* = 0.003, t–x—*p* < 0.001, y—*p* = 0.04.

**Figure 6 nutrients-16-02568-f006:**
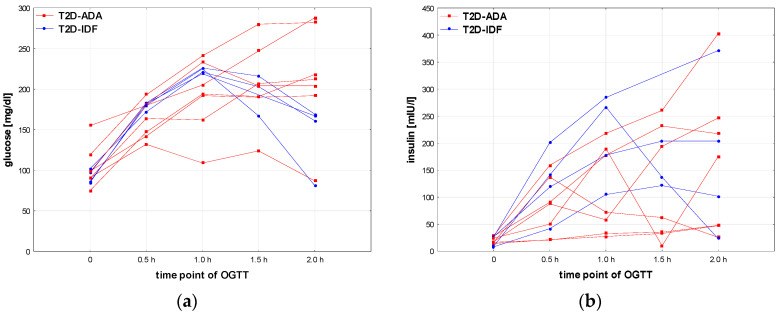
Glucose (**a**) and insulin (**b**) profiles of particular patients with T2D diagnosed according to the criteria of ADA [8] or IDF only [13].

**Table 1 nutrients-16-02568-t001:** Auxological characteristics of the study group and of the subgroups diagnosed before and after COVID-19 pandemic (PRE and POST).

	All	PRE	POST	*p* *
N (m/f)	263 (134/129)	113 (59/54)	150 (75/75)	
Age [years]	13.6 (11.5–15.9)	13.5 (11.7–16.1)	13.4 (11.1–15.8)	0.782
Height [cm]	165.0 (155.0–172.0)	165.0 (155.0–172.0)	165.0 (154.0–173.0)	0.578
Weight [kg]	87.1 (69.8–105.0)	85.1 (71.0–96.7)	89.0 (69.0–108.6)	0.223
BMI [kg/m^2^]	31.2 (28.5–35.2)	30.8 (28.6–33.2)	32.0 (28.0–37.0)	0.085
BMI z-score	2.39 (2.11–2.69)	2.31 (2.07–2.53)	2.49 (2.15–2.87)	0.056

* *p*-values relate to the differences between groups PRE and POST in Mann–Whitney U-test.

**Table 2 nutrients-16-02568-t002:** Indices of insulin sensitivity, lipid profile, and other biochemical tests in the study group and in the subgroups diagnosed before and after COVID-19 pandemic (PRE and POST).

	All	PRE	POST	*p* *
HOMA	3.35 (2.17–5.05)	3.54 (2.46–5.05)	3.14 (2.12–5.05)	0.159
QUICKI	0.32 (0.31–0.34)	0.32 (0.30–0.33)	0.33 (0.31–0.34)	**0.006**
Matsuda index	2.71 (1.90–3.78)	2.59 (1.72–3.67)	2.88 (2.08–4.09)	**0.044**
AUC (ins/glu)	0.94 (0.62–1.26)	0.98 (0.65–1.55)	0.87 (0.60–1.15)	**0.003**
AUC-glu	255 (231–280)	250 (224–278)	258 (233–282)	0.090
AUC-ins	235 (153–333)	245 (161–378)	221 (148–314)	0.114
HbA1c [%]	5.4 (5.2–5.6)	5.4 (5.2–5.6)	5.4 (5.2–5.6)	0.544
T-Ch [mg/dL]	160 (142–180)	165 (146–188)	156.5 (138–178)	**0.026**
HDL [mg/dL]	43 (37–49)	43.5 (37–49)	43 (37–49)	0.667
LDL [mg/dL]	108 (91–132)	106 (85–125)	110.5 (96–134)	0.081
TG [mg/dL]	101 (78–139)	101.5 (85–142)	101 (74–137)	0.221
Uric acid [mg/dL]	6.20 (5.38–7.12)	6.34 (5.50–7.25)	6.05 (5.21–7.06)	0.227
Creatinine [mg/dL]	0.60 (0.50–0.71)	0.59 (0.49–0.70)	0.61 (0.52–0.71)	0.362
ALT [IU/L]	23 (18–31)	24 (16–35)	23 (19–29)	0.680
AST [IU/L]	23.5 (17–31)	22.5 (18–29)	25 (16–34)	0.852

* *p*-values relate to the differences between groups PRE and POST in Mann–Whitney U-test. Significant differences (*p* < 0.05 are marked in bold).

**Table 3 nutrients-16-02568-t003:** Auxological characteristics of patients with T2D, prediabetes, and NGT, diagnosed according to the criteria of ADA.

	NGT	Prediabetes	T2D	*p* *
N (m/f)	182 (89/93)	74 (41/33)	7 (4/3)	
Age [years]	13.5 (11.3–16.2)	13.6 (11.8–15.8)	15.2 (11.9–15.8)	0.761
Height [cm]	165 (155–173)	165 (155–172)	165 (161.5–177)	0.771
Weight [kg]	86.8 (68.4–104.0)	87.4 (74.2–105.0)	98.5 (80.0–108.0)	0.622
BMI [kg/m^2^]	31.1 (28.1–35.5)	32.1 (29.0–34.8)	31.2 (29.4–37.8)	0.556
BMI z-score	2.37 (2.07–2.70)	2.44 (2.19–2.64)	2.55 (2.01–2.97)	0.565

* *p*-values relate to the differences between groups T2D, prediabetes, and NGT in Kruskal–Wallis test.

**Table 4 nutrients-16-02568-t004:** Indices of insulin sensitivity, lipid profile, and other biochemical tests in the patients with T2D, prediabetes, and NGT, diagnosed according to the criteria of ADA.

	NGT	Prediabetes	T2D	*p* *
HOMA	3.16 (2.01–4.73)	3.61 (2.47–5.52)	5.31 (3.93–5.81)	**0.002**
QUICKI	0.32 (0.31–0.34)	0.32 (0.30–0.33)	0.30 (0.30–0.31)	**<0.001**
Matsuda index	3.07 (2.08–4.35)	2.19 (1.47–2.91)	1.76 (1.22–2.73)	**<0.001**
AUC (ins/glu)	0.98 (0.59–1.23)	1.01 (0.75–1.28)	1.01 (0.75–1.28)	0.054
AUC-glu	244 (223–266)	284 (257–311)	284 (257–311)	**<0.001**
AUC-ins	212 (140–314)	279 (207–382)	279 (207–382)	**<0.001**
HbA1c [%]	5.3 (5.2–5.5)	5.6 (5.4–5.8)	6.5 (5.5–8.0)	**<0.001**
T-Ch [mg/dL]	157 (140–176)	157 (140–176)	185 (165–204)	**0.009**
HDL [mg/dL]	43.5 (38–49)	43.5 (38–49)	46 (38–52)	0.254
LDL [mg/dL]	106 (84–125)	106 (84–125)	139 (116–173)	**0.002**
TG [mg/dL]	96 (74–125)	96 (74–125)	153 (114–169)	**<0.001**
Uric acid [mg/dL]	6.05 (5.21–7.06)	6.55 (5.54–7.30)	7.22 (6.22–9.70)	**0.022**
Creatinine [mg/dL]	0.61 (0.50–0.70)	0.60 (0.49–0.71)	0.72 (0.54–0.80)	0.189
ALT [IU/L]	22 (18–29)	24 (18–40)	22.5 (28–27)	0.195
AST [IU/L]	22 (17–29)	25 (17–35)	46 (18–65)	**0.068**

* *p*-values relate to the differences between groups T2D, prediabetes, and NGT in Kruskal–Wallis test. Significant differences (*p* < 0.05 are marked in bold).

**Table 5 nutrients-16-02568-t005:** Auxological characteristics of patients with T2D, prediabetes, IH-1h, and NGT, diagnosed according to the criteria of IDF.

	NGT-IDF	IH-1h	Prediabetes-IDF	T2D-IDF	*p* *
N (m/f)	144 (65/79)	37 (23/14)	71 (40/31)	11 (6/5)	
Age [years]	13.4 (11.1–16.1)	14.1 (11.9–16.7)	13.5 (11.4–15.8)	14.4 (12.9–15.8)	0.678
Height [cm]	163.5 (154.5–170)	166 (155–175)	165.5 (155–172)	165 (161.5–177)	0.551
Weight [kg]	84.4 (69.0–104.0)	91.0 (67.4–113.0)	86.3 (74.0–106.4)	98.5 (82.0–105.0)	0.488
BMI [kg/m^2^]	31.1 (28.1–35.0)	31.8 (27.4–36.8)	32.0 (29.0–34.6)	32.1 (29.4–37.8)	0.705
BMI z-score	2.37 (2.09–2.68)	2.39 (2.05–2.84)	2.42 (2.19–2.62)	2.55 (2.01–2.97)	0.727

* *p*-values relate to the differences between groups T2D-IDF, prediabetes-IDF, IH-1h, and NGT-IDF in Kruskal–Wallis test.

**Table 6 nutrients-16-02568-t006:** Indices of insulin sensitivity, lipid profile, and other biochemical tests in the patients with T2D, prediabetes, IH-1h, and NGT, diagnosed according to the criteria of IDF.

	NGT-IDF	IH-1h	Prediabetes-IDF	T2D-IDF	*p* *
HOMA	3.28 (2.10–4.90)	2.93 (1.95–4.70)	3.60 (2.47–5.40)	5.31 (3.82–5.81)	**0.011**
QUICKI	0.32 (0.31–0.34)	0.33 (0.32–0.35)	0.32 (0.30–0.33)	0.30 (0.30–0.31)	**0.003**
Matsuda index	3.07 (2.08–4.55)	3.08 (1.99–3.60)	2.21 (1.51–2.91)	1.76 (1.20–2.75)	**<0.001**
AUC (ins/glu)	0.89 (0.60–1.23)	0.76 (0.55–1.27)	1.01 (0.75–1.30)	0.88 (0.47–1.13)	0.102
AUC-glu	235 (220–254)	273 (267–284)	283 (257–306)	353 (335–385)	**<0.001**
AUC-ins	210 (134–298)	214 (153–346)	279 (207–382)	278 (178–410)	**0.002**
HbA1c [%]	5.3 (5.1–5.5)	5.35 (5.2–5.5)	5.6 (5.4–5.8)	5.6 (5.4–6.7)	**<0.001**
T-Ch [mg/dL]	157 (141–177)	152.5 (130–175)	166 (144–194)	170 (157–192)	0.063
HDL [mg/dL]	44 (38–50)	43 (37–49)	40 (37–49)	40 (35–50)	0.442
LDL [mg/dL]	106 (86–125)	108 (78–129)	113.5 (98–141)	121 (106–144)	**0.013**
TG [mg/dL]	96 (74–127)	96 (69–124)	123.5 (84–177)	152 (114–169)	**0.001**
Uric acid [mg/dL]	6.03 (5.15–7.06)	6.05 (5.30–7.09)	6.58 (5.50–7.30)	7.06 (6.05–8.90)	0.070
Creatinine [mg/dL]	0.60 (0.50–0.70)	0.63 (0.49–0.73)	0.58 (0.48–0.71)	0.68 (0.54–0.77)	0.262
ALT [IU/L]	22 (17–28)	26 (21–35)	24 (18–40)	22.5 (18–33)	**0.028**
AST [IU/L]	22 (17–28)	25 (16–41)	25 (17–35)	46 (18–65)	**0.038**

* *p*-values relate to the differences between groups T2D-IDF, prediabetes-IDF, IH-1h, and NGT-IDF in Kruskal–Wallis test. Significant differences (*p* < 0.05 are marked in bold).

## Data Availability

The raw data supporting the conclusions of this article will be made available by the authors on request; further inquiries can be directed to the corresponding author.

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
