# Peer review of "Obesity-Related Complications Including Dysglycemia Based on 1-h Post-Load Plasma Glucose in Children and Adolescents Screened before and after COVID-19 Pandemic"

_nutrients, 2024, doi:10.3390/nu16152568_

Round 1

Reviewer 1 Report

Comments and Suggestions for Authors

Comments to the authors

After carefully reviewing this paper, edtitled "Obesity-related complications including dysglycemia based on 1-hour post-load plasma glucose in children and adolescents screened before and after COVID-19 pandemic" by Joanna Smyczyńska et al., Some concerns could arise that may help improve this paper.

1: The background section fails to highlight the novelty of the study. Many similar studies have been conducted regarding the impact of the COVID-19 pandemic on obesity and metabolic disorders. The introduction could benefit from a clearer focus on the specific objectives and hypotheses of the study.

2: The retrospective, observational design limits the ability to infer causality. A longitudinal or controlled study design would provide more robust evidence. The description of methods, particularly the OGTT and the criteria used for categorizing glucose metabolism disorders, lacks sufficient detail. This makes it difficult to assess the rigor and reproducibility of the study. The paper uses non-parametric tests but does not justify this choice. Additionally, the paper does not explain why specific post-hoc tests were chosen, which could question the validity of the results.

3: The results section relies heavily on p-values without discussing the clinical significance of the findings. This can be misleading, as statistically significant results are not always clinically important. The tables (such as Table 1 and Table 2) are overly complex and difficult to interpret. They contain too much information, some of which is irrelevant to the primary objectives. The paper would benefit from more visual aids like graphs or charts to illustrate key findings more clearly and make the data more accessible to readers.

4: The discussion does not adequately interpret the findings within the broader context of existing literature. The authors should compare their results more extensively with previous studies. There is a lack of discussion on potential biological mechanisms underlying the observed changes in glucose metabolism and insulin sensitivity post-COVID-19. Some conclusions are speculative and not sufficiently supported by the data presented. For example, the assertion that the COVID-19 pandemic has caused an increase in metabolic disorders is not backed by robust evidence from this study alone.

5: The limitations section is brief and does not fully acknowledge significant limitations such as potential biases in the retrospective design, the lack of a control group, and the generalizability of the findings. The study does not adequately address potential confounding factors that could influence the results, such as variations in diet, physical activity, or socio-economic status during the pandemic.

6: The writing style is verbose and could be more concise. This would enhance readability and comprehension. There are inconsistencies in the use of terminology and statistical reporting throughout the paper.

Overall, while the study addresses an important public health issue, it has several major weaknesses and areas that need improvement to enhance its scientific rigor and impact.

Comments on the Quality of English Language

Moderate editing of English language required

Author Response

Dear Reviewer,

thank you for very detailed and inspiring comments and suggestions. The authors would like to provide that we tried to do our best to improve the text according to these recommendations.

Comment 1: The background section fails to highlight the novelty of the study. Many similar studies have been conducted regarding the impact of the COVID-19 pandemic on obesity and metabolic disorders. The introduction could benefit from a clearer focus on the specific objectives and hypotheses of the study.

Response 1:

Data on previous reports implementing the IDF criteria in children have been added in "Introduction". We’ve corrected the description of IDF diagnostic criteria that in previous version of the manuscript was really somewhat inaccurate (see lines 80-88 of current version of the manuscript).

Comment 2: The retrospective, observational design limits the ability to infer causality. A longitudinal or controlled study design would provide more robust evidence. The description of methods, particularly the OGTT and the criteria used for categorizing glucose metabolism disorders, lacks sufficient detail. This makes it difficult to assess the rigor and reproducibility of the study. The paper uses non-parametric tests but does not justify this choice. Additionally, the paper does not explain why specific post-hoc tests were chosen, which could question the validity of the results.

Response 2:

The authors are aware of the limitations of retrospective, observational studies. Nevertheless, COVID-19 pandemic started in 2020, while IDF recommendations are from 2024, so, they could not be implemented in prospective studies including pre-pandemic period. Pandemic COVID-19 was an unexpected event, thus prospective projects including its effect on public health including pre-pandemic period were rather impossible to plan. 

OGTT criteria have been described in more detail  (see lines  135-138 of current version of the manuscript) and additionally included in "Graphical abstract" which is added according to the suggestion of other Reviewer. We’ve also corrected the description of IDF diagnostic criteria that in previous version of the manuscript was really somewhat inaccurate (see lines 80-88 of current version of the manuscript).

The information  concerning the distribution of particular variables has been provided in the final part of the section "Materials and Methods".  Before starting the actual analysis, the distribution of all the variables in particular groups has been assessed in Kolmogorov-Smirnov test. As the normality was violated for the majority of variables, we decided to use nonparametric tests. Dunn test was chosen for post-hoc comparison as appropriate for the groups of various sizes.

Comment 3: The results section relies heavily on p-values without discussing the clinical significance of the findings. This can be misleading, as statistically significant results are not always clinically important. The tables (such as Table 1 and Table 2) are overly complex and difficult to interpret. They contain too much information, some of which is irrelevant to the primary objectives. The paper would benefit from more visual aids like graphs or charts to illustrate key findings more clearly and make the data more accessible to readers.

Response 3:

The issue of the results obtained in our study that are statistically significant with no clinical importance has been briefly commented in “Discussion” (lines 344-347 of present version of the manuscript).

The authors appreciate the importance of graphical presentation of research results, but they tried to maintain a balance between the number of Tables and Figures in the manuscript. Moreover, the data in Tables can be presented with greater accuracy (impossible to read from the size of the columns or whiskers), while adding the exact numerical values ​​of medians and interquartile ranges in the Figures for several groups in two time intervals would not be more readable than just providing them in Tables. We hope that "Graphical abstract" makes tracking the results of the study easier. 

Comment 4: The discussion does not adequately interpret the findings within the broader context of existing literature. The authors should compare their results more extensively with previous studies. There is a lack of discussion on potential biological mechanisms underlying the observed changes in glucose metabolism and insulin sensitivity post-COVID-19. Some conclusions are speculative and not sufficiently supported by the data presented. For example, the assertion that the COVID-19 pandemic has caused an increase in metabolic disorders is not backed by robust evidence from this study alone.

Response 4:

The discussion has been extended by adding an additional review of existing literature concerning the effect of COVID-19 pandemic on development of disorders of carbohydrate metabolism, related to increased rate of obesity  (see green highlights in text). We’ve pointed at the need of further studies on possible direct links between these phenomena.

Comment 5: The limitations section is brief and does not fully acknowledge significant limitations such as potential biases in the retrospective design, the lack of a control group, and the generalizability of the findings. The study does not adequately address potential confounding factors that could influence the results, such as variations in diet, physical activity, or socio-economic status during the pandemic.

Response 5: Additional information concerning potential biases of our study has been included (see blue highlight in Discussion” section – lines 444-449).

Comment 6: The writing style is verbose and could be more concise. This would enhance readability and comprehension. There are inconsistencies in the use of terminology and statistical reporting throughout the paper.

Response 6: The paper has been carefully read by experienced researcher in order to improve language inaccuracies (the changes of the text are highlighted in yellow).

Thank you once again for all valuable advices.

Kind regards,

Authors

Reviewer 2 Report

Comments and Suggestions for Authors

The manuscript presented by Smyczyńska et al., is very interesting. Relating obesity and metabolic alterations to COVID-19 is important in nutrition and health (clinical and public). The manuscript is well written, the methodology is appropriate and sufficient. The results support the discussion. However, I have the following (minor) comments.

I. Comments:

1. Improve the writing of the objective of the study.

2. The pandemic generated by COVID-19 was controlled (importance of vaccines). However, we could face an epidemic or pandemic. In this regard, I suggest briefly discussing the projections of this study considering that obesity generates an increase (sub-clinical) in the inflammatory response. People with obesity are more susceptible to viral complications, particularly due to a greater predisposition to an excessive inflammatory response.

3. It would be interesting to propose the relevance of a healthy diet to face the problem.

Author Response

Dear Reviewer,

thank tou for the review and suggestions. According to your comments, the following corrections and additions have been made:

Comment 1: Improve the writing of the objective of the study

Response 1: The sentence is re-written as follows: "The aim of the study was to assess the occurrence of carbohydrate metabolism disorders and to compare selected indices of IS/IR and of other metabolic complications in children and adolescents with simple obesity ..." (see green highlight in text - line 103 of present version of the manuscript)

Comment 2: The pandemic generated by COVID-19 was controlled (importance of vaccines). However, we could face an epidemic or pandemic. In this regard, I suggest briefly discussing the projections of this study considering that obesity generates an increase (sub-clinical) in the inflammatory response. People with obesity are more susceptible to viral complications, particularly due to a greater predisposition to an excessive inflammatory response.

Response 2: That's very important issue. "Discussion" section has been extended by including references related to these problems (see green highlights).

Comment 3: It would be interesting to propose the relevance of a healthy diet to face the problem.

Response 3: Some information about the relevance of diet, undertaken and proposed interventions has been added in "Discussion" (see green highlights)

Kind regards,

Authors

Reviewer 3 Report

Comments and Suggestions for Authors

I read the article intitulated Obesity-related complications including dysglycemia based on 1-hour post-load plasma glucose in children and adolescents screened before and after COVID-19 pandemic.

The article can be improved by a graphical abstract.

Please detail the inclusion and exclusion criteria to include patients in the study

Please note the limitation of the study.

Author Response

Dear Reviewer,

Thank you for the review and suggestions. Please find our responses below.

Comments 1: The article can be improved by a graphical abstract

Response 1: Graphical abstract is added

Comment 2: Please detail the inclusion and exclusion criteria to include patients in the study

Response 2: Incusion and exclusion criteria are explained in more detail (see blue highlights in "Materials and Methods" section - lines 119-123 of present version of the manuscript)

Comment 3: Please note the limitation of the study

Response 3: Appropriate paragraph is added in section "Discussion" (lines 444-449 of present version of the manuscrpt, highlighted in blue).

Kind regards,

Authors

Round 2

Reviewer 1 Report

Comments and Suggestions for Authors

The authors have addressed all the cocnerns